# Insufficient Physical Fitness and Deficits in Basic Eating Habits in Normal-Weight Obese Children Are Apparent from Pre-School Age or Sooner

**DOI:** 10.3390/nu13103464

**Published:** 2021-09-29

**Authors:** Martin Musálek, Petr Sedlak, Hana Dvořáková, Anna Vážná, Jan Novák, Jakub Kokštejn, Šárka Vokounová, Adéla Beránková, Jana Pařízková

**Affiliations:** 1Faculty of Physical Education and Sport, Charles University, José Martího 31 Veleslavín, 162 52 Praha 6, Czech Republic; jakubkokstejn@seznam.cz (J.K.); vokounova@ftvs.cuni.cz (Š.V.); berankova02@gmail.com (A.B.); 2Faculty of Science, Charles University, Albertov 6, 128 00 Praha 2, Czech Republic; petr.sedlak-uk-prf@seznam.cz (P.S.); annamvazna@gmail.com (A.V.); jan.novak@natur.cuni.cz (J.N.); 3Pedagogical Faculty, Charles University, Magdalény Rettigové 47/4, 110 00 Praha 1, Czech Republic; hannadvorak@seznam.cz (H.D.); parizkova.jan@gmail.com (J.P.)

**Keywords:** normal-weight obesity, preschool age children, eating habits, physical fitness

## Abstract

Normal-weight obesity appears to be an extended diagnosis/syndrome associated with insufficient physical fitness levels and inadequate eating habits at least from school years. However, its relation to long term health parameters in pre-school children remains unknown, even though pre-school age is crucial for the determining healthy lifelong habits. Therefore, the aim of the current study was to investigate the differences in physical fitness level and basic eating habits between normal-weight obese, normal-weight non-obese, and overweight and obese preschoolers. The research sample consisted of 188 preschoolers aged 4.0–6.9 years (M_age_ = 5.52 ± 0.8 year), normal-weight obese = 25; normal-weight non-obese = 143, overweight and obese = 20. Body composition was measured using bio-impedance InBody230. Six tests assessed the physical fitness level: sit-ups; standing long jump; shuttle running 4 × 5 meters; throwing with a tennis ball; multistage fitness tests; sit and reach. A four-item eating habits questionnaire for parents focusing on breakfast regularity, consumption of sweet foods and drinks, selection of food and attitude towards eating was used. A non-parametric analysis of variance and Fisher’s exact test along with suitable effect sizes were used for data processing of physical fitness tests and the basic eating habits questionnaire, respectively. Normal-weight obese children performed significantly worse (from *p* = 0.03 to *p* < 0.001, ES ω2-G = low to medium) in muscular fitness, cardiorespiratory fitness and running agility compared to normal-weight non-obese counterparts and did not significantly differ in the majority of physical fitness performance tests from overweight and obese peers. In basic eating habits, normal-weight obese boys preferred significantly more sweet foods and drinks (*p* = 0.003 ES = 0.35, large), while normal-weight obese girls had significantly more negative attitude towards eating (*p* = 0.002 ES = 0.33, large) in comparison to their normal-weight non-obese peers. Normal-weight obesity seems to develop from early childhood and is associated with low physical fitness and deficits in eating habits which might inhibit the natural necessity for physically active life from pre-school age or sooner.

## 1. Introduction

Normal-weight obesity, characterized by an excessive amount of body fat along with normal body mass index (BMI), negatively affects human health [1]. Even though recent research has pointed out that normal-weight obesity is related to nutrition or eating deficiencies and physical performance deficits, its causes and the period of its development still remain unclear [2]. Previous studies in human adults reported that normal-weight obese individuals face similar metabolic, cardiovascular, and inflammation health risks [3] as evidently obese (high body fat, high BMI) people [4,5,6], they consume significantly smaller amounts of antioxidants and have poor physical fitness [7]. Physical fitness (PF) level is described as physical readiness of the organism manifested in muscle strength, endurance, speed, or movement coordination [7], and from childhood is significantly associated with long term physical health [8], obesity prevalence [9], and cognitive development [10,11]. The low level of PF in childhood is usually related to higher metabolic risks, obesity [12], low levels of physical activity (PA) and worse performance in cognitive tests [13]. Although the pre-school age period was found as crucial for long term lifestyle habit determination including eating [14], and PF has been recognized as an important health parameter [15,16], the association between normal-weight obesity, PF and eating habits in pre-school age has been unknown. Therefore, the aim of the current study was to investigate the differences in physical fitness level and basic eating habits between normal-weight obese (NWO), normal-weight non-obese (NWNO), and overweight and obese (Ow&Ob) preschoolers. Sedlak et al. [17,18] repeatedly pointed at secular changes (longitudinal over generation changes) of the increasing amount of body fat and the decreasing amount of lean mass on extremities in different samples of Czech preschoolers, which has a connection to the development of normal-weight obesity [19]. Musalek et al. [20] found that NWO pre-school children performed significantly worse in fundamental movement skills and had more than three times higher prevalence for sever motor difficulties compared to NWNO peers. Studies conducted in school children and adolescents found that NWO individuals have a decreased amount of lean mass on extremities, a more fragile skeleton [21], higher risk of metabolic and cardiovascular problems [22,23] and poor cardiorespiratory fitness, muscular strength and agility [24,25,26]. From an eating habits perspective previous research pointed out that NWO adolescents tend to omit breakfast [24] and that NWO adults seem to consume much fewer dietary antioxidants (they have significantly lower intakes of root vegetables, cereals and fish) and a higher intake of confectionery monosaccharides, [7,27]. Most studies of the effect of dietary fat intake on weight in children find a paradoxical small to no effect [28], but it does not mean that there are no changes in body build. In this context, one recent animal study [29], found that an early high fat diet in juvenile rats applied for nine weeks caused a significant increase of body fat without an excessive increase of body weight with a low ratio between the muscle mass and body weight compared to normally fed rats. In human such data are not available. This condition is considered as normal weight obesity, and in rats it seemed to be stable across their life span. In addition, in humans, normal-weight obesity was also found to be a stable diagnosis in girls from middle school age to early adulthood [22]. From the aforementioned findings, we therefore hypothesize that normal-weight obesity is negatively associated with PF level and basic eating habits (regularity of breakfast, consumption of sweet foods and drinks, the level of meal selection, attitude to eating) already in pre-school age. Furthermore, we assume that due to weak muscle development, NWO pre-school children will have similar PF level deficits as Ow&Ob counterparts.

## 2. Methods

This research study is a cross-sectional type of study. The methods and results of the current study were described in line with the recommendation of *The Strengthening the Reporting of Observational Studies in Epidemiology (STROBE) Statement: guidelines for reporting observational studies* [30]. Participants: 8 kindergartens in the Central Bohemia region and the capital Prague were randomly recruited for the current study from 2017–2019. The inclusion criteria for the participants included (1) child aged of 4.0–6.9 years; (2) no limitations in exercise participation; (3) a signed informed consent from legal representatives of all children participating in the study. At an information meeting arranged in each kindergarten parents were acquainted with the aim and process of study. Out of 308 informed consents 214 signed consents were obtained (69.5% acceptance). Six participants with neurological conditions that would affect their physical fitness performance were excluded, and 20 children with incomplete data. Therefore, the final research sample consisted of 188 preschoolers *n* = 84 boys, *n* = 104 girls in age range from 4.01 to 6.5 (mean = 5.5 ± 0.8). From these 188 children complete data from anthropometry, physical fitness, and a basic eating habits questionnaire were obtained.

The data were always obtained in the spring period of the school year April–May 2017, April–May 2018, April–May 2019, and the data collection was carried out in all kindergartens at the same time of the day, between 8:30 A.M. and 11:00 A.M. The study was approved by the institutional review board of the Faculty of Science-Charles University in Prague under approval No. 2017/23 from 2 October 2017.

### 2.1. Anthropometry

Regarding anthropometry, body height, body weight and body composition were measured.

In each kindergarten, the children were measured by four academically trained examiners in two separate rooms. These rooms complied with all requirements stipulated by Czech Decree No. 410/2005 Coll. on education of children. Children were measured in separate room with a teacher in presence.

The anthropometric measurements were conducted according to standard anthropometric techniques [31,32]. Body height was measured using a portable stadiometer Trystom (Trystom, s.r.o., Olomouc, Czech Republic; accuracy 1 mm) in a standardized upright position with the head positioned in a horizontal Frankfurt plane and defined as the distance between the vertex point and the floor. Body mass (in kg) was ascertained by a stamped weighting scale (precise to 0.1 kg). Body mass was ascertained by a stamped weighting scale (precise to 0.1 kg). Body mass index (BMI) was calculated as BMI = body mass (in kilograms)/body height (in meters)^2^.

A bio-impedance analysis (BIA) was conducted using InBody 230 and software Lokin’Body (DMS-BIA technology; InBody Co., Seoul, Korea), which the manufacturer declares to be applicable for children from the age of 3 to estimate body composition (body fat and muscle mass). The amount of body fat percentage (BF%) and the percentage of skeletal muscle mass were estimated. This instrument was validated against dual-energy X-ray absorptiometry for young school age children with satisfactory results for estimating body fat (Source: https://www.ncbi.nlm.nih.gov/pmc/articles/PMC5291432/) (accessed on 10 April 2017) [33]. Subjects were measured wearing underwear, guidelines on how to prepare a child for measurement were given in the paper form to parents and kindergarten teachers (e.g. the participant should not be extensively physically active one day, prior the measurement, they should be adequately hydrated, they should not be measured after food consumption).

For each child, fat to muscle ratio index was calculated as a ratio of skeletal muscle and body fat percentage received from bio-impedance [34]. 

### 2.2. Physical Fitness Tests

All PF tests were administrated in a gym hall with space requirements: length 22 × 12 m. Assessing of PF included six widely accepted physical fitness tests [35,36,37] which measured: (1) dynamic strength of lower limbs—standing long jump; (2) Strength endurance of trunk—sit ups; (3) dynamic strength of upper limb and trunk—overhead tennis ball throwing; (4) agility—4 × 5 m running; (5) running endurance—Multistage fitness test 20 m; and (6) flexibility—sit and reach test assessed. PF and its role and development in the preschool age period is determined in Framework Educational Programs issued by Ministry of Education Czech Republic in 2016 and 2017 (available from https://www.msmt.cz/vzdelavani/predskolni-vzdelavani/ramcovy-vzdelavaci-program-pro-pv-1?lang=1; https://www.msmt.cz/vzdelavani/predskolni-vzdelavani/ramcovy-vzdelavaci-program-pro-predskolni-vzdelavani-od-1) (accessed on 1 September 2016). Development of PF in preschool environment is not purposefully developed through PF tests, however, is arranged in the form of simple controlled as well as spontaneous games. Nevertheless, we realize that children could have previous motor experience with some of tests due to their leisure time activities spent out of preschool—kindergarten environment. 

In gym hall, children always participated in a standardized 10 min warm-up session under the supervision of one examiner. One aim was also to familiarize children via games with movements used in PF tests. After that, the children were divided into two groups which proceeded with all five PF tests in a fixed sequence. A progressive 20 m shuttle run test was always the last one done and had special rules (see below).

Each physical fitness test was explained and physically demonstrated to the children by the examiner.

Standing long jump: The length of the jump in centimeters was evaluated, and each participant was allowed three attempts. The distance was measured from the starting line to the rear edge of the last footprint.Sit-ups: Strength and endurance of the abdominal muscles was measured with the FitnessGram form of curl-ups. A pre-recorded voice with visual instruction for the children was used. Children must keep the knees bent at 140°, feet in contact with the floor and hands placed at the sides, palms down. They must come all the way “up” to the point; in the case of preschoolers their fingers travel 7.5 cm from the resting position to a line on the mat (as shown on the right) and then return with their head touching the mat on the “down” command. The number of correctly performed curl-ups on metronome cadence was then recorded [38,39].Overhead throwing with a tennis ball: Children were given a tennis ball in the preferred hand. The child had to throw the ball overhead as far as possible. The examiner demonstrated the execution of the throw, because only overhead throws were counted. Each child threw the ball three times with the preferred hand and three times with the non-preferred hand. The longest attempt was recorded.Shuttle running 4 × 5 m: Children started to run from a starting line from standing position on signal Ready-steady-GO! Each child had to run four times the distance of 5 meters which was determined by two colour cones as fast as possible. At the end of each track, child had to touch on top of colour cone and run as fast as possible back. Each child had a trial run, after which each participant had two attempts with a five-minute rest between each attempt. The time the participant needed to run the whole four tracks was recorded. The fastest time on 0.1 s was recorded.Sit and reach: The modified bench with a height of 25 cm, not 30 cm as in the original version (https://lafayetteevaluation.com/products/121086-sit-reach-box, accessed on 22 September 2021), was used for pre-school children. The child was asked to sit down against the wall, straighten the lower limbs and the back and lean against the wall with the whole back. A bench was moved to the legs (feet) so that the entire surface of the child’s feet rested on one side of the bench. Subsequently, the child was asked to stretch the arms forward. The metric scale on the bench was moved to the tip of the middle finger. Then the child performed a forward bend. While bending, the child must not flex the knees of the lower limbs. The examiner checked this by holding his hand on the child’s knees throughout the whole trial. The maximum distance the child reached without breaking the rules was recorded. The child performed the entire test twice, in rapid succession.Multistage fitness test: Children repeatedly ran the distance of 20 m within the designated area. The aim was to keep progressively increasing the speed of the running on the track for as long as possible. In this study a version for pre-school children according to [40] with the starting velocity of 6.5 km/h was used. Four children ran in one group. The speed was determined by the examiner, who ran a little ahead of the children. The second examiner ran slightly behind the children. The number of completed shuttles of each child was recorded, then converted into completion time [37].

### 2.3. Questionnaire, Eating Habits

In this study a four-item questionnaire for parents, monitoring the basic eating habits of their children was used see Table 1. This questionnaire represents a standard screening protocol used in the Czech pre-school and school environment and was internally validated in cooperation with the Czech State Institute for Health and Institute of Endocrinology (http://www.szu.cz/index.php?lang=2; https://www.endo.cz/en/ (accessed on 1 January 2017).

This questionnaire was always administered in paper form to each parent at a kindergarten meeting along with text and a verbal explanation related to each question.

### 2.4. Definition of Normal-Weight Obese, Overweight and Obese and Normal Weight Non-Obese Categories of Children

To define normal-weight obese, normal-weight non-obese and overweight and obese children, we used Czech reference values of body fat percentage for pre-school children [41,42] and Czech BMI charts for pre-school children respecting the International Obesity Task Force (IOTF) cut-offs for thinness, overweightness and obesity [43]. In the first step we divided the children according to their body fat percentage into two groups. The results achieved by each child from InBody230 were compared with Czech reference values for body fat in pre-school age. The children were divided into two groups:(1)Children with the non-excessive amount of body fat <90th centile of body fat of Czech reference values for each age category.(2)Children with the excessive amount of body fat >90th centile of body fat of Czech reference values for each age category.

Along with body fat percentage, the body mass index was calculated for each child. We used BMI IOTF standards when a child in range BMI 15th–85th was classified as normostenic/normal-weight non-obese; BMI > 85th overweight and obese. In our sample there were no children with a BMI lower than the 15th centile.

According to the following rules the children were divided into three categories:(1)Normal-weight non-obese: BMI in range 15th–85th and with body fat percentage <90th centile of Czech reference values(2)Normal-weight obese children with BMI 15th–85th and with body fat percentage >90th centile of Czech reference values(3)Overweight and obese BMI > 85th and body fat percentage > 90th centile of Czech reference values

The research sample consisted of *n* = 25 normal-weight obese (NWO), *n* = 143 normal-weight non-obese (NWNO) and *n* = 20 overweight and obese (Ow&Ob) children see Table 2, mean age 5.52 ± 0.8.

Even though the proportion of children in the defined categories is highly unbalanced, this ratio between NWO and NWNO children corresponds with previous findings of Czech sample studies Kopecky 2011 [20,21,26,44], and with international studies [45]. Further, the ratio between NWNO and Ow&Ob children in our sample corresponds with the prevalence of overweightness and obesity in Czech preschoolers [46]. 

### 2.5. Statistical Analysis

Even though all PF tests are commonly used for scientific purposes, there had not been any international or national Czech reference values considering age and sex for pre-school age. Therefore, we converted all raw scores from PF tests to a common Z-score scale considering the age and sex of the participants. Shapiro Wilk and Anderson Darling tests rejected in seven out of nine assessed parameters because of data normality. Therefore, the Kruskal-Wallis analysis of variance (ANOVA) was conducted. Multiple regression analysis was used to study relationships between continuous variables (body composition, results of PF tests). Further, the Dunn’s post-hoc test to further explore the sources of the differences was used. Statistical significance was accepted at *p* < 0.05 along with effect size parameter generalized Hays ω2 (ω2-G) [47] with cut-offs small effect <0.059; medium effect 0.059–0.137; and large effect > 0.137 [48]. Fischer’s test and contingency tables were used for the analysis of results from the eating habits questionnaire as a suitable method for smaller sample sizes. Statistical significance was accepted at *p* < 0.05 along with effect size estimated using Cramér’s V [49]. The data were analyzed in R-CRAN environment, The jamovi project (2021). jamovi. (Version 1.6) [Computer Software] and NCSS2007 [50,51,52]. Within the manuscript we present data using average and standard deviation form for easy interpretation.

## 3. Results

Children from all three groups considering sex did not significantly differ in their chronological age and body height, even though Ow&OB participants were approximate-ly two centimeters taller than their counterparts from other groups in both sexes. Further, Ow&OB children were significantly heavier and had significantly higher BMI and body fat% with a large effect size compared to both NWO and NWNO peers. NWO participants did not differ in weight and BMI compared to NWNO counterparts; according to the study design, they had a significantly higher amount of body fat% than NWNO children. On the other hand, NWO and Ow&Ob children had significantly less muscle mass than NWNO see Table 3. Therefore, when we compared the fat to muscle ratio of children from the three defined groups, the NWO and Ow&Ob preschoolers did not differ in this parameter; how-ever, both had significantly worse fat to muscle ratio than NWNO counterparts. Since the development of muscle and fat mass is sex-dependent, we transformed fat to muscle ratio values to Z-scores.

In PF tests, NWO children performed significantly worse in muscle strength endurance of the trunk (sit-ups); cardiorespiratory fitness (Multistage fitness test); the dynamic strength of lower limbs (standing long jump); and running agility (shuttle run 4 × 5 m) compared to their NWNO peers see Table 4. However, only in cardiorespiratory fitness, dynamic strength of lower limbs and running agility was the size effect at medium level. No significant differences were found in overhead throwing with the right and left upper limb. Furthermore, Dunn’s post hoc tests showed that in muscle strength endurance of the trunk, cardiorespiratory fitness and running agility, NWO children performed not significantly better than Ow&OB children. In dynamic strength of lower limbs, Ow&Ob children performed significantly worse compared to their NWO and NWNO peers.

In the Appendix A
Table A1, we also provide values in raw units (centimeters, seconds, or number of executions) considering group status: (1) NWO; (2) NWNO; (3) Ow&Ob.

Furthermore, we analyzed which of the anthropometric parameters are significantly associated with physical performance. Body weight, body height, percentage of body fat, and fat to muscle ratio were used as independent variables. Multiple regressions showed that body height is positively associated with dynamic muscle strength of lower limbs, cardiorespiratory level, and agility running. Fat to muscle ratio was positively associated with the dynamic strength of lower limbs and agility running. Percentage of body fat was negatively related to dynamic strength of the lower limbs, cardiorespiratory fitness, and agility running. Body weight was negatively associated with cardiorespiratory fitness. Dynamic strength of lower limbs is significantly positively associated with body height and muscle fat ratio and negatively associated with the amount of body fat (R^2^ = 0.41). Cardiorespiratory fitness level was found to be significantly positively related to body height and negatively to amount of body fat (R^2^ = 0.26) see Table 5. 

The results from the eating habits questionnaire analyzed with Fisher’s exact test brought two fundamental findings, Table 6. The first is that NWO children regardless sex did not significantly differ in the regularity of having breakfast (*p* = 0.26, Cramér’s V = 0.1), in the degree of meal selection when eating (*p* = 0.44, Cramér’s V = 0.09); and even in the preference for consumption of sweet foods) and drinks (*p* = 0.282, Cramér’s V = 0.05 in comparison to their peers. The second finding is that NWO children significantly vary in different preference for consumption of sweet foods and drinks when considering sex. While NWO and Ow&Ob boys were 22 and 14 times more likely, calculated as ODDS ratio, to consume sweet foods and drinks, the NWO girls did not differ in their preference for sweet foods and drink consumption in comparison to their NWNO counterparts. Further, the likelihood for consumption of sweet foods and drinks between NWO and Ow&Ob boys was not significantly different *p* = 0.8, Confidence Interval CI95% = 0.01–33.5. In girls, Ow&Ob preschoolers did not differ in preference of consuming sweet food and drinks from NWNO girls; however, they were 27 times less likely to be highly selective in food than NWNO peers. 

The question, *Does your child eat almost everything,* however, does not give us information about the process of eating. Therefore, the extended question was formulated related to the child’s attitude to food, asking whether a child is disobedient, not finishing the portion of food or if a child needs an extra portion of a meal. Since the question *What is your child’s attitude to food?* has three response choices, we provide for better orientation results of this question in separate Table 7. In regard to attitude to eating, we found that NWO girls became significantly more non-compliant during food consumption—they did not want to eat compared to their NWNO and Ow&Ob counterparts. Moreover, Ow&Ob girls usually asked for extra portions. 

## 4. Discussion

### 4.1. Anthropometry and Body Differences between NWO and NWNO Preschoolers

From an anthropometric perspective, NWO children had a significantly greater amount of body fat, and less amount of muscle mass along with non-different BMI compared to NWNO peers. Therefore, NWO preschoolers also had a significantly lower fat to muscle ratio than NWNO children which was on average close to the results of Ow&Ob children. These findings are in conformity with previous findings revealed in school children. In [21], focused on muscle mass surface on limbs, we found that NWO school children in the age range 8–10.9 had significantly decreased the muscle mass surface on upper and particularly on lower extremities in comparison to NWNO children. Low muscle mass and high amount of body fat have also been connected with several health problems. Wiklund et al. [22], in their longitudinal study revealed that NWO girls who had significantly lower muscle /fat ratio compared to their NWNO peers also had significantly higher cardiometabolic risk. In adults instead of previously mentioned the high fat to muscle ratio has been associated with metabolic syndrome or insulin resistance [53,54,55]. 

### 4.2. Differences in Physical Fitness Level between NOW, Ow&Ob, and NWNO Preschoolers

In PF level NWO as well as Ow&Ob children performed significantly poorer in muscle strength endurance of trunk (sit-ups), dynamic strength of lower limbs (standing long jump), cardiorespiratory fitness (multistage fitness test) and agility (shuttle 4 × 5 m) compared to NWNO counterparts. These results correspond with our previous findings [26], where NWO school children also performed significantly poorer in all the above mentioned PF areas compared to NWNO peers. Furthermore, same as in our previous study [26], NWO preschoolers performed as badly as Ow&Ob counterparts in running agility, dynamic strength of lower limb, and cardiorespiratory fitness. However, the following question araised. Whether the amount of body fat, body weight, body height, or muscle fat/ratio influence the most performance in PF tests. Results of multiple regressions showed that body height was the strongest predictor for performance in dynamics strength of lower limbs, cardiorespiratory fitness, and running agility. Fat to muscle ratio was negatively related to performance in dynamic strength of lower limbs and running agility. On the other hand, the percentage of body fat was negatively associated with dynamic strength of lower limbs, cardiorespiratory fitness, and agility running. Since the NWO children were the shortest in stature and Ow&Ob were the tallest, the influence of body height does not seem to be reasonable. Both performed poorly in a majority of PF tests. Therefore, we suggest that an excessive amount of body fat in combination with low muscle development is possible explanation for weak PF performance of NWO and Ow&Ob preschool children. This suggestion is in conformity with outcomes [56] and [21] which found negative association of body fatness and physical fitness performance from preschool age period, and pointed out that NWO as well as Ow&Ob school children showed the largest muscle development deficits on lower limbs in comparison to NWNO peers. In addition, muscle fitness was recognized to be highly positively connected to muscle mass development and negatively associated with fat mass and cardiometabolic problems [16]. Therefore, in our opinion the lean mass development and particularly on limbs fat to muscle ratio seems to be significantly associated with PF from early childhood. In addition, the PF level is significantly correlated with qualitative aspects of human movement such as fundamental movement skills [57] which involve balance, manipulative activities and locomotor activities. These FMS are often assessed in pre-school age period because FMS represent building blocks of further movement development including PF [58]. Our results therefore also support our findings preschoolers [20], where NWO pre-school children performed significantly worse in the level of fundamental movement skills compared to NWNO peers. Considering the fact, that previous research on pre-school age samples revealed a significant connection between the level of PF, the level of fundamental movement skills and physical activity [59,60,61], we can assume that NWO children as well as Ow&Ob children might be physically less active than their NWNO peers. This assumption arises from the results of previous studies on preschool age children that found that the amount of fat-free mass is negatively associated with sedentary behavior and positively associated with performance in PF [62]. Even though until today no study has been done that would investigate physical activity regime of NWO preschoolers, recent studies have repeatedly shown that obese pre-school children who have a low fat-free mass index and a high fat mass index tend to be less physically active compared to NWNO peers with greater evidence in boys [63,64]. In addition, only one study focused on physical activity in NWO, Olafsdottir et al. [24] found that Icelandic NWO adolescents are less physically active compared to their NWNO peers. 

### 4.3. Differences in Basic Eating Habits between NWO, Ow&Ob, and NWNO Preschoolers and Their Connection to Physical Fitness

In the context of eating habits, we found that NWO children did not differ in the regularity of breakfast and selection of meals compared to NWNO peers. However, significant differences between NWNO and NWO preschoolers were revealed in sweet foods and drinks consumption preferences. Moreover, these differences were not uniform when considering sex. While NWO and Ow&Ob boys prefer significantly more frequent consumption of sweet foods and drinks NWO girls did not differ from NWNO girl peers. Since boys are naturally more physically active than girls from early childhood [65,66] we assume that there could exist a link between a preference for sweet foods and drinks consumption and motivation for physical activity and PF level. A possible connection between normal-weight obesity assumed poor PF and possible decreased amount of physical activity of NWO individuals, described in the previous paragraph was suggested by [67]. In this study, children with normal weight but with metabolic abnormalities showed to have a significantly higher monosaccharides intake, a high endotoxin level associated with systematic health disease along with a significantly lower physical activity regime compared to NWNO counterparts. Regular consumption of sweet confectionery in the long term is associated with increasing of pro-inflammatory markers such as C-reactive protein (CRP), interleukin 6 (IL-6) or tumor necrosis alpha (TNF alpha) [68]. In animal studies, a high-saccharide diet led to disruption of barrier integrity, greater gut permeability with a subsequent increasing level of pro-inflammatory markers IL-6, TNF alpha [69]. Further, in human studies, the elevated level of IL-6 or C-reactive protein was positively associated with physical inactivity and negatively related to the amount of leisure time dedicated to physical activity in children and adults [70,71,72]. Therefore, we assume that the recurring preference for the consumption of sweet foods and drinks in boys might inhibit the spontaneous necessity for physical activity in children, which is naturally higher in boys than girls from early childhood. This assumption is also based on the results of brain imaging studies which showed that sweets consumption directly decreases the activity of (1) insula, co-operating in motor control, perception and emotion; (2) cingulate cortex, responsible for the results of motivation transformed to behavioral patterns, and (3) basal ganglia which are responsible for movement initiation and are connected to the development of motor cortex [73]. On the other hand, there is the question of why NWO girls, who did not prefer to consume sweet foods and drinks, achieved the same poor PF level as NWO boys. It seems that physical inactivity of NWO children is common for boys and girls; however, the causes of this inactivity might be different considering sex. In boys, they might be significantly more led by the increased preference for sweet foods and drinks consumption. While in girls it might be caused by a rather negative attitude to eating, the implication of which could be a low caloric intake without the demand for spontaneous physical activity. Further, for the following research we would highly recommend to collect data about daily food composition which might address the question, whether development of normal weight obesity in girls is significantly influenced with diet like high-fat diet which was found in rats [28]. In our opinion, the different preferences in sweet foods and drinks consumption of NWO girls and boys seem to be highly important because if we had analyzed the data without considering sex, no differences would have been revealed.

#### Limitations

The fundamental strength of the present study is the homogeneous research sample, which represented the common Czech population of pre-school children. A further strength of this study is that all measured variables were collected in the same calendar year period. PF and eating habits might differ based on climate changes during the year. On the other hand, we realize that even though the ratio of NWO, NWNO and Ow&Ob children represents a common ratio in the population, the first view on unbalanced samples with low frequency of participants in the NWO and Ow&Ob categories might evoke limitations of the present study. Moreover, deeper information about the physical habits, particularly physical activity monitoring and the social environment of families involved in the study, could explain the eating habit differences between the sexes that was revealed. Finally, the present study was cross-sectional in design, and, therefore, causality cannot be discerned.

## 5. Conclusions

NWO preschoolers had low fat to muscle ratio and showed significant deficits in variety aspects of physical fitness compared to NWNO. In addition, NWO boys prefer significantly more sweet foods and sweet drinks consumption, while NWO girls have a greater negative attitude towards food consumption. Both could however inhibit the natural necessity for physical activity, which is positively associated with PF of children. Since NWO preschoolers already exhibit serious physical and eating deficits, future studies should focus on the identification of markers which lead in the early stages of human life to the development of normal-weight obesity in children. 

## Figures and Tables

**Table 1 nutrients-13-03464-t001:** Czech screening questionnaire for parents monitoring basic eating habits of their children.

Item	Option for Response
Does your child eat breakfast every day?	yes/no
Does your child eat almost everything?	yes/no
Is your child keen on sweet foods and drinks?	yes/no
What is your child’s attitude to food?	Eats rather almost anything they are given and portion is enough Usually does not want to eat what is served, is disobedient during eating, served portion of food does not finish Wants more food after finishing the portion

**Table 2 nutrients-13-03464-t002:** Frequency distribution of children from *n* = 188 in each defined category.

Category	Boys	Girls	Boys & Girls
Normal-weight obese (NWO)	13	12	25
Normal-weight non-obese (NWNO)	65	78	143
Overweight and obese (Ow&Ob)	7	13	20

**Table 3 nutrients-13-03464-t003:** Anthropometry description of NOW, NWNO and Ow&Ob children.

Item	NWNO	NWO	Ow&OB	Chi-Square	*p*-Value	ω2-G
Age in years	5.5 ± 0.8	5.56 ± 0.9	5.45 ± 0.8	0.19	0.83	0.001
Height in cm	114 ± 7.2	113.1 ± 5.3	115.8 ± 8.9	0.83	0.42	0.001
Weight in kg	19.4 ± 2.7	20.1 ± 2.6	26 ± 4.9	28.4	<0.001 (a)	0.28
BMI	14.9 ± 0.9	15.3 ± 0.8	19.1 ± 1.9	66.4	<0.001 (a)	0.62
Body fat %	14.8 ± 3.1	23.8 ± 3.1	30.4 ± 4.3	99.7	<0.001 (b)	0.73
Muscle mass %	39.8 ± 2.8	35.1 ± 6.9	33.8 ± 10.4	45.0	<0.001 (c)	0.36
Fat to muscle ratio Z-score	−0.40 ± 0.4	0.58 ± 0.45	2.33 ± 1.1	88.5	<0.001 (c)	0.67

The data are presented as mean ± SD. Letters (a), (b), (c) marked which group or groups significantly differed at level *p* < 0.001. (a) Difference between NWNO and Ow&Ob, NWO and Ow&Ob groups; no significant difference between NWO and NWNO; (b) Difference between all groups: Ow&Ob and NWO, Ow&Ob and NWNO, NWO and NWNO groups; (c) Difference between NWNO and Ow&Ob, NWNO and NWO, no significant difference between NWO and OWOB. NWNO = normal-weight non-obese; NWO = normal-weight obese; Ow&Ob = overweight and obese; BMI = body mass index.

**Table 4 nutrients-13-03464-t004:** Physical fitness performance of NWO NWNO and Ow&Ob children considering sex.

Item	NWNO	NWO	Ow&OB	Chi-Square	*p*-Value	ω2-G
Standing long jump	0.29 ± 0.9	−0.24 ± 0.9	−0.83 ± 1.0	21.7	<0.001 (a)	0.11
Sit-ups	0.13 ± 1.1	−0.33 ± 0.6	−0.27 ± 0.9	7.1	0.03 (c)	0.03
Throw right	0.03 ± 1.1	−0.11 ± 0.8 (b)	−0.18 ± 0.9 (b)	0.38	0.82	0.001
Throw left	0.08 ± 1.1	0.04 ± 1.0	−0.12 ± 0.8	0.60	0.74	0.001
Shuttle 4 × 5 m	0.27 ± 0.8	−0.28 ± 0.6	−0.53 ± 1.0	15.1	<0.001 (c)	0.07
Sit and reach	−0.01 ± 1.1	−0.36 ± 1.0	0.12 ± 1.1	2.12	0.34	0.001
Multistage fitness test	0.23 ± 1.0	−0.26 ± 0.7	−0.53 ± 1.2	15.5	<0.001 (c)	0.07

The data are presented as mean ± SDLetters (a), (b), (c) marked which group or groups significantly differed at level *p* < 0.001. (a) Difference between NWNO and Ow&Ob, NWO and Ow&Ob groups; no significant difference between NWO and NWNO; (b) Difference between all groups: Ow&Ob and NWO, Ow&Ob and NWNO, NWO and NWNO groups; (c) Difference between NWNO and Ow&Ob, NWNO and NWO, no significant difference between NWO and OWOB; NWNO = normal-weight non-obese; NWO = normal-weight obese; Ow&Ob = overweight and obese; BMI = body mass index.

**Table 5 nutrients-13-03464-t005:** The role of anthropometric parameters in physical performances, multiple regression models.

Model	Predictors	Regression Coefficient	*t*-Value	*p*-Value	R^2^
Standing long jump	Body height	1.51	3.9	<0.001	0.41
	Fat to muscle ratio	3.99	−2.1	0.03	
	Percentage body fat	−1.3	−4.01	<0.001	
Shuttle 4 × 5 m	Body height	5.9	4.7	<0.001	0.31
	Fat to muscle ratio	4.13	2.06	0.03	
	Percentage body fat	−1.94	2.23	0.02	
Multistage fitness test	Body height	5.7	4.53	<0.001	0.26
	Body weight	−4.9	−2.13	0.03	
	Percentage of body fat	−1.8	−1.98	0.04	

**Table 6 nutrients-13-03464-t006:** Frequencies of responses in dichotomously scored items from the Czech screening questionnaire for parents monitoring basic eating habits.

	BOYS *n* = 84				GIRLS *n* = 104			
Breakfast	Yes	No	*p*	Effect Size (Cramér’s V)	Yes	No	*p*	Effect Size (Cramér’s V)
NWO	11	1	0.02	0.35	12	1	0.52	0.11
NWNO	64	1			75	4		
Ow&OB	5	2			12	1		
Sweet consumption								
NWO	12	0	0.003	0.35	8	5	0.46	0.11
NWNO	34	31			38	41		
Ow&OB	6	1			5	8		
Eats everything								
NWO	6	6	0.80	0.06	9	4	0.09	0.22
NWNO	36	29			39	40		
Ow&OB	3	4			10	3		

Breakfast: Does your child eat breakfast every day? Sweet consumption: Is your child keen on sweet foods and drinks? Eat everything: Does your child eat almost everything?

**Table 7 nutrients-13-03464-t007:** Frequencies of responses in polytomous scored items from Czech screening questionnaire for parents monitoring basic eating habits.

		BOYS *n* = 84				GIRLS *n* = 104				
What Is Your Child’s Attitude to Food?	Usually Is Disobedient, Not Finishing the Portion	Eats Whole Portion, Usually Anything They Are Given	Wants More Food after Finishing the Portion	*p*	Effect Size (Cramér’s V)	Usually is Disobedient, Not Finishing the Portion	Eats Whole Portion, Usually Anything They Are Given	Wants More Food after Finishing the Portion	*p*	Effect Size (Cramér’s V)
NWO	4	8	1	0.17	0.23	10	12	0	0.03	0.25
NWNO	16	44	5			27	47	5		
Ow&OB	2	2	3			2	8	3		

## Data Availability

Data supporting reported results can be found at https://www.researchgate.net/publication/354871931_Data_Sheet_normal-weight_obesity.

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
