# Peer review of "Insufficient Physical Fitness and Deficits in Basic Eating Habits in Normal-Weight Obese Children Are Apparent from Pre-School Age or Sooner"

_nutrients, 2021, doi:10.3390/nu13103464_

Round 1

Reviewer 1 Report

Dear Authors,

Congratulation for a nice and interesting article, some small point have to be clarified and we made some suggestions that we hope will contribute to a even better discussion. We wish you success in the publication.  

Line 97, please put the identification acceptance of the review board

One possible limit for the study: did the children experimented the tests before?

Line 143, fitnessGram sit up please put Reference for this test in preschool children

Lines 210 and 212 correct fate

Line 223 check the size of “children” characters

Why not assessed the physical activity? Maybe it is a clearly a limit for the study interpretation!

Table 4. We suggest also putting the results in cm, second, number of executions and not only in Z score. For many lecturers it should be of interest. For the item please use the same words or abbreviation than in the methods and put the unity of the measures.

Table 5 is not in conformity with table 1. In table 1, What is your child’s attitude to food? Have 3 choices. In the table 5 there is only one option. After in table 6 you do another treatment Please clarify this in methods and use uniform terms (Naughty in table 6 in not in methods).

Table 5 comments,

What about p 0.02 in breakfast in boys?  (See also what you write in line 362 and 363)

When you write: “NWO children regardless sex did not significantly differ in regularity of having breakfast, and in the degree of meal selection when eating” where is it supported in table 5?              “NWO preschoolers have significantly different preference for consumption of sweet foods and drinks in comparison to NWNO peers.” where is it supported in table 5?     

What is the relation between muscle/fat ratio and the results in the tests were the weight have influence on the performance like agility, board jump and also shuttle leger test? To give some arguments to your lines 338 and 339 discussion, you should study this with your data’s. It should be better than the arguments of lines 340 to 342 that are complementary but not essentials for this discussion. You have the data’s to show this relation (correlation, regression).

Lines 344 to 346 please specify which test permit you to make this affirmation, we suppose that it is on the basis of the agility and throwing test results? By the way line 419 to 421 seems to be in contradiction with lines 344 to 346?

Lines 366 about the differences between NWO girls and boys about sweets consumption, if the girls consume significantly less sweets they are NWO like the boys, difference in PA could be a reason and you have to consult the studies about PA differences in preschool boys and girls NOWO vs Overweight and obesity to discuss better this aspect.  Additionally, maybe the NWO girls consume more fats than the NWO boys? Like it is suggested in study 27 with rats it can be a reason for NWO. May be some studies you have consulted with NWO subject can help you to discuss this idea?

Author Response

First of all, thank you very much for your valuable review we appreciate it a lot! All our changes and corrections are highlighted with yellow colour.

Line 97, please put the identification acceptance of the review board

Response: added into manuscript

One possible limit for the study: did the children experimented the tests before?

Response: added into manuscript

Line 143, fitnessGram sit up please put Reference for this test in preschool children

Response: references added into manuscript

Lines 210 and 212 correct fate

Response: corrected

Line 223 check the size of “children” characters

Response: corrected

Why not assessed the physical activity? Maybe it is a clearly a limit for the study interpretation!

Response: We did two pilot data collections with GT9X accelerometers with kindergarten children in co-operation with their parents. Unfortunately, the responsibility of parents for supervision of wearing accelerometers was very low and final data contains only results from part of normostenic children. We absolutely realize that this is main limit for results interpretation. We emphasized it in part of Limit of study.

Table 4. We suggest also putting the results in cm, second, number of executions and not only in Z score. For many lecturers it should be of interest. For the item please use the same words or abbreviation than in the methods and put the unity of the measures.

Response:  we tried to make this table however, raw units would have to be present for each age category and each sex. Therefore, we left table 4 with Z-score values, but we added Appendix table with raw values. It is presented beyond of reference list.

Table 5 is not in conformity with table 1. In table 1, What is your child’s attitude to food? Have 3 choices. In the table 5 there is only one option. After in table 6 you do another treatment Please clarify this in methods and use uniform terms (Naughty in table 6 in not in methods).

Response: thank you very much it was really confusing. Therefore we explained better where is the conceptual difference between these two questions – Eating everything and Attitude to food. Attitude to food is extended question of Eating everything.

Table 5 comments,

What about p 0.02 in breakfast in boys?  (See also what you write in line 362 and 363)

Response: Thank you very much. We added complete information about differences in eating habits when analyzing boys and girls together.

What is the relation between muscle/fat ratio and the results in the tests were the weight have influence on the performance like agility, board jump and also shuttle leger test? To give some arguments to your lines 338 and 339 discussion, you should study this with your data’s. It should be better than the arguments of lines 340 to 342 that are complementary but not essentials for this discussion. You have the data’s to show this relation (correlation, regression).

Response: we did multiple regression to address which of anthropometry is predictor significantly associated with PF performance.

Lines 344 to 346 please specify which test permit you to make this affirmation, we suppose that it is on the basis of the agility and throwing test results?

Response: we adjusted this section, we provided more detail information in this section of discussion.

By the way line 419 to 421 seems to be in contradiction with lines 344 to 346?

Response: there are two different tests: muscle endurance strength of trunk  was assess with sit-ups however, throwing represents dynamic strength of trunk and upper limb

Lines 366 about the differences between NWO girls and boys about sweets consumption, if the girls consume significantly less sweets they are NWO like the boys, difference in PA could be a reason and you have to consult the studies about PA differences in preschool boys and girls NOWO vs Overweight and obesity to discuss better this aspect.  

Response: this information is in manuscript between lines 421 – 433.

Additionally, maybe the NWO girls consume more fats than the NWO boys? Like it is suggested in study 27 with rats it can be a reason for NWO. May be some studies you have consulted with NWO subject can help you to discuss this idea?

Response: it might reason, however, we have no data about this. Our evidence is that NWO girls are “naught” during food consumption these girls are leaving meal on plate so they do not finish portion. Now it is a very good question why. However, if I do not regularly finish my food, from whatever reason I have lower caloric intake, than if I have lower caloric intake I should have lower amount of fuel for PA. May they are not interested in PA they just have different priorities, because previous thesis at our university suggested that NWO children in adolescent age are significantly higher day dreamers.

Reviewer 2 Report

Thank you so much for inviting me to review this article. Authors tried to to find whether normal-weight obesity in pre-school age is already connected with worse degree of physical fitness and bad eating habits. The article deals with an interesting topic in Czech children. The article uses appropriate statistical procedures. It also details very well all the procedures used. I enjoyed reading it very much. Congratulations to the authors for their effort. 

Minor comments:

- Authors should include the aim of the study in the introduction. 

- Please, change "bad" eating habits by "inadequate" eating habits. 

- Lines 307-308 "The aim of this study was to find whether normal-weight obesity in pre-school age 307 is already connected with worse degree of PF and bad eating habits." Authors could eliminate this information (it is repetitive).

- Please, try to reduce the conclusion section (it is very extensive).

Best wishes,

Author Response

First of all thank you very much for valuable review we appreciate it a lot!

- Authors should include the aim of the study in the introduction. 

Response: added into Introduction section

- Please, change "bad" eating habits by "inadequate" eating habits. 

Response: corrected

- Lines 307-308 "The aim of this study was to find whether normal-weight obesity in pre-school age 307 is already connected with worse degree of PF and bad eating habits." Authors could eliminate this information (it is repetitive).

Response: we took out this sentence

- Please, try to reduce the conclusion section (it is very extensive).

Response: we took out redundant information

Reviewer 3 Report

Title:  Weak physical fitness and deficits in basic eating habits in normal-weight obese children are apparent form pre-school age or sooner

In this manuscript the authors investigated the differences in physcal fitness and basic eating habits between normal weight obese , normal weight non obese e overweight obese preschoolers. The authors conclude that normal weight obesity is associated with low physical fitness and deficits in eating habits. The topic is interesting and important and the methods used seem adequate however, especially the description of the method section on the eating habits questionnaire and the results from this questionnaire is very confusing. In addition the discussions section must be more concise and English language needs extensive editing  limitations and strengths.

Strengths of the study:

The subject studied in the manuscript is of interest. The objective of the study is clear. It is important to study the pre-school population

Limitations of the study  

The manuscript needs extensive rewriting, currently  there are major issues.

In more detail:

Title: consider changing "weak" to "reduced"?

Abstract:

The abstract should be more concise. The first sentence states insufficient physical fitness and eating habits. I suggest to define what the authors consider to be insufficient. 

Introduction 

Line 38 "along with" = combined with 

Line 44 "great body fat" is high body fat?

Rewrite sentence 45 to 48 physical fitness to development

Line 54 secular changes? please explain

Lie 56- 57 which has a connection to the development of normal weight obesity, please add a  reference

Line 67 legumes. Is eating legumes not good? please explain or delete comment because it creates confusion

Line 68: I suggest removing the animal study and add arguments/findings from human studies

Line 74 and 77 "assume" do you mean hypothesize?

Methods

Line 90 " neurologic anamnesis" do you mean neurologic diseases affecting physical fitness?

Line 112 repeat "body mass - kg"

Line 134 "passed -same" = participated in a standardized

Line 156-161 Please rewrite to be more clear

Line 164 What original version?

Table 1 "eats - portion" Are these 3 answer possibilities?

Line 207.  delate "non obese". If I understood it correctly, the definition of obese depends on the body fat%

Statistical analysis: I do not consider myself to have enough knowledge to evaluate the statistical analysis used.

Results:

Line 243 to 246 "Ow OB  to peers" Add " according to study design"

Line 247 " however" change to in according to the study design"

Line 281 to 303 describe the result of the diet habit questionnaire. This is a confusing part and should be rewritten. Maybe it is a suggestion to start with a comparison between the 3 different groups including p values and only after the whole group comparison a comparison between boys and girls. I suggest to add a whole group column to table 5

Table 5 " eats everything" from what question are these results obtained?

Line 301 " non compliance during food consumption" what do the authors mean?

Discussion

Line 308: please define "worse degree of PF and bad eating habits"

In general the discussion needs to be much more concise and rewritten. In addition, when the authors refer to their own studies I suggest they refer to their own studies in a different way. for example, " we have previously presented" 

Conclusion:

The authors repeat results in the conclusion making the conclusion more an addition to the result section 
